# Antitumor Activity of Nanoparticles Loaded with PHT-427, a Novel AKT/PDK1 Inhibitor, for the Treatment of Head and Neck Squamous Cell Carcinoma

**DOI:** 10.3390/pharmaceutics13081242

**Published:** 2021-08-12

**Authors:** Joaquín Yanes-Díaz, Raquel Palao-Suay, María Rosa Aguilar, Juan Ignacio Riestra-Ayora, Antonio Ferruelo-Alonso, Luis Rojo del Olmo, Blanca Vázquez-Lasa, Ricardo Sanz-Fernández, Carolina Sánchez-Rodríguez

**Affiliations:** 1Department Otolaryngology, Hospital Universitario de Getafe, Getafe (Madrid), Carretera de Toledo, km 12.500, 28905 Madrid, Spain; joaquin.yanes@salud.madrid.org (J.Y.-D.); juanignacio.riestra@salud.madrid.org (J.I.R.-A.); rsanzf@salud.madrid.org (R.S.-F.); 2Department of Polymeric Nanomaterials and Biomaterials Institute of Polymer Science and Technology CSIC, C/Juan de la Cierva, 3, 28006 Madrid, Spain; rpsuay@ictp.csic.es (R.P.-S.); mraguilar@ictp.csic.es (M.R.A.); rojodelolmo@ictp.csic.es (L.R.d.O.); bvazquez@ictp.csic.es (B.V.-L.); 3Biomedical Research Centre in Bioengineering Biomaterials, and Nanomedicine CIBER-BBN, 28029 Madrid, Spain; 4Department of Medicine, Faculty of Biomedical and Health Sciences, Universidad Europea de Madrid, Villaviciosa de Odón, 28670 Madrid, Spain; 5Fundación de Investigación Biomédica del Hospital Universitario de Getafe, Carretera de Toledo, km 12.500, 28905 Madrid, Spain; antoniojose.ferruelo@salud.madrid.org; 6CIBER de Enfermedades Respiratorias, Instituto de Investigación Carlos III, 28029 Madrid, Spain

**Keywords:** PHT-427 inhibitor, polymeric nanoparticle, PI3K pathway, AKT, PDK1, hypopharynx carcinoma squamous cells

## Abstract

Currently, new treatments are required to supplement the current standard of care for head and neck squamous cell carcinoma (HNSCC). The phosphatidylinositol3-kinase (PI3K) signaling pathway is commonly altered and activated in HNSCC. PHT-427 is a dual PI3K-mammalian target of the AKT/PDK1 inhibitor; however, to the best of our knowledge, the effect of the PHT-427 inhibitor on HNSCC has not been investigated. This study aims to evaluate the antitumoral effect of PHT-427-loaded polymeric nanoparticles based on α-tocopheryl succinate (α-TOS). The in vitro activity of PHT-427 was tested in hypopharynx carcinoma squamous cells (FaDu) to measure the cell viability, PI3KCA/AKT/PDK1 gene expression, and PI3KCA/AKT/PDK1 levels. Apoptosis, epidermal growth factor receptor (EGFR), and reactive oxygen species (ROS) were also measured. The presence of PHT-427 significantly enhances its antiproliferative and proapoptotic activity by inactivating the PI3K/AKT/PDK1 pathway. Nanoparticles (NPs) effectively suppress AKT/PDK1 expression. Additionally, NPs loaded with PHT-427 produce high oxidative stress levels that induce apoptosis. In conclusion, these results are promising in the use of this nanoformulation as a PHT-427 delivery system for effective HNSCC treatment.

## 1. Introduction

Of all cancer cases worldwide, head and neck squamous cell carcinomas (HNSCC) account for 5% [1]. Despite advances in diagnosis and treatment, HNSCC presents a poor prognosis, with a 5-year survival rate of 50–60% [1,2,3,4]. Therefore, novel agents that significantly enhance the effects of existing chemotherapeutic drugs with lower toxicity are needed.

In terms of genetic aberrations, a promising pathway in HNSCC is the phosphatidyl-inositol 3-kinase (PI3K) signaling pathway. The PI3K signaling pathway regulates cell proliferation, cell survival, and apoptosis [5,6]. Genetic aberration or dysregulation of the genetic components of the PI3K signaling pathway, including AKT, PDK1, PTEN, and PIK3CA, has been found to play an important role in multiple aspects, including uncontrolled proliferation, angiogenesis, resistance to apoptosis, and metastatic capability [7].

Defects in the PI3K pathway are found in a variety of human cancers [8], including HNSCC [9]. Most abnormalities in the PI3K signaling pathway occur through mutations and amplified oncogenes of the PI3K catalytic subunit alpha isoform (PIK3CA), or mutations or loss of the mixed-function lipid phosphatase tumor suppressor PTEN [10]. In addition, AKT can be oncogenic if the pleckstrin homology (PH) domain is mutated, increasing its affinity for PI3-phosphates, and consequently leading to increased phosphorylation at the cellular membrane [10]. For these reasons, the development and use of pharmacologic inhibitors targeting PI3K, AKT, and mTOR for cancers such as HNSCC in preclinical and clinical studies has surged [11]. However, in our study, PH domains of the proteins AKT and PDK1 were inhibited. PHT-427 (4-dodecyl-N-(5-(5-(methyl(7-nitrobenzo(c)(1,2,5)oxadiazol-4-yl)amino)pentyl-1,3,4-thiadiazol-2-yl)benzenesulfonamide) binds to the PH domains of both AKT and PDPK1 and inhibits the activity of both proteins in pancreatic, prostate, ovarian, breast, and non-small cell lung cancer cells and in tumor xenografts of pancreatic cancer [11,12] but has never been tested in HNSCC. Despite all of these properties, the main obstacle to the successful application of PHT-427 is the hydrophobic nature of the drug, which significantly reduces its bioavailability and therapeutic activity.

Efforts to eliminate these problems have focused on developing new drug delivery systems such as microspheres, microparticles, and nanoparticles [13]. Polymeric nanoparticles have successfully addressed the problems related to drug delivery, preventing drug degradation, increasing drug bioavailability, and decreasing the toxic effects of the drug [14,15,16,17,18]. In this sense, Lucero-Acuña et al. [19] encapsulated the PHT-427 inhibitor into poly(lactic-co-glycolic) acid (PLGA) nanoparticles. PHT-427-loaded NPs improved the antitumor effect of PHT-427 in pancreatic cancer models. Similarly, our group is devoted to synthesizing nanovehicles of the methacrylic derivative of α-tocopheryl succinate (MTOS) and to copolymerizing amphiphilic macromolecules that can self-assemble in aqueous media, forming surfactant-free nanoparticles (NPs). The hydrophilic segment is formed by N-vinyl pyrrolidone (VP), and the hydrophobic segment is formed by MTOS: poly(VP-*co*-MTOS) (89:11) [20,21]. We have recently demonstrated that poly(VP-*co*-MTOS) nanoparticles loaded with α-tocopheryl succinate (α-TOS) had antitumor and antiangiogenic effects in hypopharynx squamous cell carcinoma (FaDu cells) [20,21].

We hypothesized that encapsulating PHT-427 into MTOS-based NPs to form NP-PHT427 would improve the therapeutic effect of this treatment in an HNSCC in vitro model.

## 2. Materials and Methods

### 2.1. Synthesis and Characterization of Polymeric Nanoparticles

Polymeric nanoparticles were successfully prepared using a MTOS11 polymer that was previously synthesized by free radical polymerization using N-vinyl pyrrolidone (VP) and a methacrylic derivative of α-TOS (MTOS) with a copolymer molar composition of VP: MTOS 89:11 [20,21]. These self-assembled NPs were prepared by the nanoprecipitation method. MTOS11 (10 mg/mL) and PHT-427 (10% *w*/*w* respect to the polymer) were dissolved in dioxane and incorporated drop by drop over an aqueous phase (NaCl, 100 mM) under constant magnetic stirring to obtain a polymer concentration of 2.0 mg/mL. After the nanoprecipitation, the NP suspensions were purified by dialysis for 72 h in order to remove the organic solvent and non-encapsulated PHT-427. The unloaded NPs (NP-Ø) were synthesized by the same procedure without incorporation of the drug.

To calculate the encapsulation efficiency, NP-PHT427 was freeze-dried to eliminate the aqueous phase. Afterward, PHT-427 entrapped in the NPs was quantified by absorbance spectroscopy. Specifically, NP-PHT427 (5 mg/mL) was dissolved in ethanol, and their absorbance was measured using a NanoDrop™ One/OneC. A calibration curve was previously calculated at PHT-427 concentrations ranging from 0.001 to 1 mg/mL in ethanol. The encapsulation efficiency (EE) was calculated using the formula below:(1)$$\mathrm{Encapsulation}\;\mathrm{efficiency}\;\left(\mathrm{EE}\right)=\frac{{\left[\mathrm{loaded}\;\mathrm{PHT}-427\right]}_{i}}{{\left[\mathrm{loaded}\;\mathrm{PHT}-427\right]}_{0}}\times 100$$
where [loaded PHT-427]_i_ is the total concentration of the PHT-427 entrapped into NPs and [loaded PHT-427]_0_ is the concentration of the molecule added initially during the preparation of NPs.

The particle size distribution of the unloaded and loaded NP suspensions was measured by dynamic light scattering (DLS) on a Malvern Nanosizer NanoZS Instrument (Malvern Panalytical Ltd., Malvern, UK) (laser wavelength 633 nm at 25 °C) and in square polystyrene cuvettes (SARSTEDT, Nümbrecht, Germany). Moreover, the scattering was monitored at a fixed angle of 173°. The zeta potential was determined for NP formulations at a 0.1-mg/mL concentration containing 10 mM of NaCl and using laser Doppler electrophoresis (LDE).

### 2.2. Cell Culture and Treatment

Hypopharynx carcinoma squamous cells (FaDu) acquired from American Type Culture Collection (ATCC, Manassas, VA, United States) were cultured in Dulbecco’s modified Eagle’s medium (DMEM) with 10% fetal bovine serum (FBS; Gibco) and 1% penicillin/streptomycin/amphotericin (Sigma-Aldrich, San Luis, CA, United States), and incubated at 37 °C and 5% CO_2_.

FaDu cells were treated with unloaded nanoparticles (NP-Ø) and loaded with the PHT-427 inhibitor (NP-427) at the concentrations of 0.5, 0.75, and 1 mg/mL for 24 or 48 h.

### 2.3. Cell Viability Assay

The cell viability was measured using Alamar Blue assay (AB) (ThermoFisher Scientific, Waltham, MA, United States). In brief, the FaDu cells were seeded at 10^4^ cells/well in 96-well plates. At 24 h of incubation, the medium was replaced with the corresponding NP treatments for 24 and 48 h. Then, AB solution (10% (*v*/*v*) solution of AB dye) was added to each well, and fluorescence was measured at 3 h of incubation with a FLUOstar Omega (BMG Labtech, Ortenberg, Germany) plate reader at an excitation wavelength of 571 nm and an emission of 601 nm. NP treatment at 0 mg/mL was taken as 100% viability. The results were averaged over 3 different independent experiments, with 16 replicates for each experimental condition per experiment.

The IC50 values were calculated using the web-based tool ATT Bioquest Inc. (https://www.aatbio.com/tools/ic50-calculator-v1, accessed on 9 August 2021).

### 2.4. Western Blot

At the end of the experiments, the FaDu cells were lysed in a buffer T-Per (Pierce, Appleton, WI, United States) with Complete™ Protease Inhibitor Cocktail (Roche, Basel, Switzerland). The cellular proteins were separated by SDS-PAGE and transferred to a nitrocellulose membrane (Bio-Rad, Hercules, CA, United States). The primary antibodies used in this study included EGFR (1/1000), p-AKT-T308 (1/5000), total AKT (1/5000), p-AKT-S473 (1/5000), and β-actin (1/2000, used as a loading control) from Abcam and phospho-PDK1-Ser241 (1/1000) and anti-PI3CA (1/1000) from Invitrogen. The immunoreactive bands were visualized using SuperSignal West Pico Chemiluminescent Substrate (ThermoFisher Scientific, Waltham, MA, United States). A densitometric analysis of the film was performed using AlphaFaseFC software (Alpha Innotech Corporation, San Leandro, CA, United States).

### 2.5. Indirect Immunofluorescence

After treatment with NP-Ø and NP-427, the FaDu seeded in 24-well plates were fixed with 4% paraformaldehyde (PFA) for 10 min. The cells were blocked for 1 h at 37 °C and incubated overnight at 4 °C with several antibodies: p-AKT-Ser473 (1/75), p-AKT-Thr308 (1/75), and Annexin-V (1/100) from Abcam (Abcam, Cambridge, UK) and p-PDK1-Ser241 and anti-PI3CA (1/100) from Invitrogen (Invitrogen Corporation y Applied Biosystems Inc., Carlsbad, United States). After, the cells were incubated with the secondary antibodies, i.e., Alexa Fluor 546-conjugated goat anti-rabbit antibody (1/250; Molecular Probes, Eugene, United States) or Alexa Fluor 488-conjugated goat anti-rabbit antibody (1/250; Molecular Probes, Eugene, United States). at 37 °C for 45 min. Then, the cells were stained with 300 nM of 4′,6-diamidino-2-phenylindole dihydrochloride (DAPI; Sigma-Aldrich, San Luis, CA, United States), reactive with fluorescent blue, which intercalates with DNA, for 5 min at 37 °C. Fluorescence was visualized using an Olympus BX51 microscope. The specificity was assessed by omitting the primary antibody. ImageJ, a free image-processing program, was used for quantitative analysis of the image.

### 2.6. Reactive Oxygen Species Detection

ROS generation was determined using red fluorescent probe dihydroethidium (DHE) (Calbiochem, San Diego, CA, United States). In brief, the FaDu cells were seeded at 1 × 10^5^ cells/well in 24-well plates. After 24 h of incubation, the cells were treated and the plates were incubated for 24 h. After fixation with 4% PFA, the cells were incubated with DHE (4 µmol/L) at 37 °C for 90 min. Next, the cells were stained with 300 nM of DAPI (Sigma-Aldrich, San Luis, CA, United States) for 5 min at 37 °C and were scored in an Olympus BX51 fluorescence microscope. ImageJ was used for quantitative analysis of the image.

### 2.7. Evaluating Reference Gene Expression

A web-based tool RefFinder (www.leonxie.com/referencegene.php, accessed on 9 August 2021) was used to determine the reference genes’ stability. RefFinder is a web-based tool developed for evaluating and screening reference genes from experimental datasets. It integrates major computational programs including BestKeeper, geNorm, Normfinder, and the comparative Delta-Ct method to rank and compare the candidate reference genes. Based on the rankings from each program, it assigned an appropriate weight to an individual gene and calculated the geometric mean of their weights for the overall final ranking [22].

The following human reference genes were tested in this study: human glucuronidase (GUSB); human ribosomal protein, large P0 (RPLP0); human transferrin receptor (p90, CD71) (TFRC); human peptidylprolyl isomerase A (cyclophilin A) (PPIA); human beta-2-microglobulin (B2M); human TATA box binding protein (TBP); human actin, beta (ACTB); and human hypoxanthine-guanine phosphoribosyl transferase I (HPRT1) (AnyGenes).

### 2.8. Quantitative Real-Time Reverse Transcription-Polymerase Chain Reaction

The total RNA derived from the FaDu cells was extracted using miRNeasy Tissue/Cells Advanced Mini Kit (QIAGEN, Hilden, Germany) following the manufacturer’s instructions. RNA recovery was determined spectrophotometrically in NanoDrop One (Thermo Scientific, Waltham, MA, United States). The total RNA (1 μg) was reverse transcribed using a StaRT kit (AnyGenes, Paris, France) according to the manufacturer’s instructions, using Veriti Thermal Cycler (Applied Biosystems, Waltham, MA, United States).

A quantitative real-time reverse transcription-polymerase chain reaction was performed to study gene expression using the 7500 Fast real-time PCR detection system (Applied Biosystems, Waltham, MA, United States). Complementary DNA (cDNA) templates (2 μL) were added to 8 μL of Perfect Master Mix SYBRG (AnyGenes, Paris, France). The final volume was 10 μL. Polymerase chain reactions were carried out according to the procedures provided by the manufacturer in triplicate. The transcript levels were normalized to GUSB and RPLPO (used as reference genes). Determination of the relative expression levels was performed using the comparative “Ct” method.

The following human primers were used in this study: PI3KCA, AKT1, AKT2, and PDK1 (AnyGenes, Paris, France).

### 2.9. Apoptosis Quantification

Apoptosis was measured by the Human Cleaved Caspase-3 (Asp175) ELISA Kit (Abcam) in cell culture extracts from each treatment group according to the manufacturer’s instructions. Absorbance was measured using a FLUOstar Omega (BMG Labtech Ortenberg, Germany) microplate reader at 450 nm.

### 2.10. Statistical Analysis

The results are expressed as mean ± standard deviation. Statistical significance (*p* < 0.05) was evaluated using the analysis of variance (ANOVA, Tukey test) by Origin 9.

## 3. Results

### 3.1. Nanoparticle Characterization: PHT-427 Encapsulation

PHT-427 was successfully entrapped in the inner core of self-assembled nanoparticles based on the MTOS11 copolymer. This copolymer was previously synthesized by free radical polymerization using VP as hydrophilic monomer and a methacrylic derivative of α-TOS (MTOS) with a molar composition of VP:MTOS 89:11. The structure of this copolymer is shown in Figure 1. Additionally, the most relevant properties of this copolymer are summarized in Appendix A (see the Appendix A).

The appropriate hydrophilic and hydrophobic balance in the polymer chains of MTOS 11 allowed for the formation of nanoparticles, where the PHT-427 was entrapped in their inner core through inter- and intramolecular hydrophobic interactions.

Unloaded (NP-Ø) and PHT-427-loaded NPs (NP-427) were successfully prepared and characterized. The results are summarized in Table 1. Both NP suspensions exhibited unimodal size distributions with an appropriate hydrodynamic diameter that slightly increased for NP-427 (132 nm and 155 nm for NP-Ø and NP-427, respectively). Additionally, PHT-427 was efficiently entrapped during the nanoprecipitation process, with an EE of 75.4% as a result of its hydrophobicity and low water solubility. In the nanoprecipitation process, the hydrophobic interactions between PHT-427 and the MTOS nanodomains in the core of the nanoparticles regulates the encapsulation of this drug. Moreover, the hydrophilic shell that was rich in VP stabilized them due to the optimal hydrophilic–hydrophobic balance in the particles.

The NPs exhibited slightly negative zeta potentials, indicating an almost neutral charge of the surface. These values are typical of α-TOS-based NPs, as recently reported by our group [15,16].

### 3.2. NPs Loaded with PHT-427 Reduce FaDu Cell Viability

The cell viability studies were assessed using an Alamar Blue assay. Figure 2 shows the effects of NP-Ø and NP-427 after 24 (A) and 48 h (B) and demonstrates that the relative FaDu viability decreased in a dose and time-dependent manner, and was lower for NP-427. Particularly, cell viability was reduced below 70%, being toxic for cancer cells (a cell viability lower than 70% is cytotoxic according to ISO 10993-5: 2009) when NP-427 was administered at concentrations higher than 0.5 mg/mL for 24 and 48 h (Figure 2A,B). The effect of NP-Ø was observed after 24 h, and this effect did not increase over time (no significant differences between 24 and 48 h were observed). The activity of NP-427 was significantly higher after 48 h; this means that the NPs were taken up by the cells at 24 h, but that the release of the PHT-427 inhibitor took longer. For instance, NP-427 at 0.5 mg/mL reduced the cell viability of FaDu from 61.5% at 24 h to 45% at 48 h.

NP-427 at concentrations of 0.5 and 0.75 mg/mL were used in the following experiments, as these concentrations were the concentrations of the NPs loaded with the incorporated highest concentration of the PHT-427 inhibitor that presented selectivity against cells at the lowest treatment time, i.e., 24 h. Moreover, NP-Ø at the concentrations of 0.5 and 0.75 mg/mL were considered to analyze the intrinsic biological activity of the polymeric nanovehicles.

The results clearly showed that NP-427 was cytotoxic for FaDu cells with an IC50 of 0.629 mg/mL at 24 h and an IC50 of 0.538 mg/mL at 48 h (see Appendix A). The total cytotoxicity (100%) was reached at a concentration of 1 mg/mL at 48 h. For NP-Ø, the IC50 is 0.738 mg/mL at 24 h, the IC50 is 0.735 mg/mL at 48 h, and the total cytotoxicity (100%) is not reached even at a concentration of 1 mg/mL. Therefore, an antiproliferative effect is seen in cancer cells. NP-427 has the lowest IC50 and, therefore, the highest efficiency.

In Appendix A (see the Appendix A), the FaDu viability when in contact with free or encapsulated PHT-427 is compared against drug concentration after 24 and 48 h of treatment. For the same drug concentration, free PHT-427 leads to a lower cell viability in contrast with the encapsulated one. After 24 h, the cell viability values were 0.047 and 0.016 mg/mL for encapsulated and free PHT-427, respectively. Therefore, NP-PHT-427 represents an excellent vehicle that allows us to encapsulate high concentrations of hydrophobic PHT-427, enhancing their bioavailability and dosage on the targeted side. In spite of the reduction in PHT-427 in vitro toxicity, NP-PHT427 allows for a sustained released of the highly encapsulated drug concentration from the inner core of the particles on the tumor area.

Figure 2C shows images obtained by optical microscopy of the FaDu culture treated with NP-Ø and NP-427 at 24 h. NP-treated groups were affected, particularly in the case of NP-427. In these groups, the cell morphology was not polygonal but rounded together with a qualitative decrease in the number of cells when compared with the control.

### 3.3. NPs Loaded with PHT-427 Induce FaDu Cell Apoptosis

Apoptosis induction by NPs in FaDu cells was quantitatively studied by measuring the levels of caspase-3 (apoptosis effector protein) by ELISA in cell lysates. The FaDu cells had significantly higher levels of caspase-3 in the NP treatment groups compared with the control group. Statistically significant differences between the levels of caspase-3 for NP-Ø and NP-427 groups were found, with the caspase-3 activity being the highest for NP-427 (Figure 3A). Apoptosis was also qualitatively analyzed by immunofluorescence using Annexin V staining cells. The fluorescence micrographs revealed an increase in red fluorescence in FaDu cells incubated with both NPs groups, being higher in the NP-427 group, indicating an increase in apoptosis (Figure 3B).

### 3.4. Reference Gene Analysis

For the selection of genes that could be used as normalizers in the qPCR analysis of the expression of target genes for FaDu cells, we analyzed eight reference gene expressions for the different experimental conditions: C (control), NP-Ø (NP without loading, 0.5 and 0.75 mg/mL), and NP-PHT427 (NP loaded with PHT427, 0.5 and 0.75 mg/mL) at 24 h.

First, the expression of the reference genes was analyzed by qPCR. Afterward, the data obtained were analyzed using the algorithm of the Red Finder software. Table 2 showed the ranking order of eight reference genes analyzed by the computational programs Normfinder, BestKeeper, Genorm, and comparative Delta-Ct method algorithms separately and recommended comprehensive ranking by the RedFinder program.

Calculations based on the algorithm of RedFinder found GUSB and RPLP0 to be the most stable genes for the NP-427 group, followed by the TFRC, PPIA, B2M, TBP, ACTB, and HPRT1 genes (Figure 4). For a good quantification of gene expression, the use of more than one reference gene for the efficient normalization of gene expression data is suggested [23]. Thus, we combined the use of the two most stable reference genes for normalization.

### 3.5. PHT-427-Loaded NPs Reduced Gene Expression of the PI3K Pathway

Quantitative statistical analysis through a histogram is used to show the real-time PCR results for PI3K pathway proteins (Figure 5). The expressions of almost every protein involved in the pathway (PI3K, AKT1, AKT2, and PDK1) were significantly (*p* < 0.05) reduced in cells treated with NP-427 in the 0.5 and 0.75 mg/mL concentrations compared with the control and NP-Ø groups.

### 3.6. Levels of PI3K Pathway Components Decreased with PHT427-Loaded NPs

The immunofluorescence identification of PI3KCA protein (green color) at 24 h with concentrations of 0.5 mg/mL and 0.75 mg/mL of NP-427 shows an objective decrease in intracellular PI3KCA (Figure 6A,B) congruent with a decrease in results in the Western blot analyses (Figure 6C) compared with untreated and NP-treated unloaded cells. This cellular decline is statistically significant (*p* < 0.05).

PDK1 identification in immunofluorescence has a similar pattern to that of PI3KCA (Figure 7A). In FaDu cultures, green fluorescence decreased in the group with NP-427, in a dose-dependent manner, compared with the control and unloaded NP groups.

The reduction in fluorescence was quantified using Image J software, and the differences found were statistically significant (*p* < 0.05) (Figure 7B). The Western blot shows the same results, with a significant decrease in PDK1 levels when the concentration of PHT427-loaded nanoparticles increased (Figure 7C). NP-427 at 0.5 and 0.75 mg/mL exhibits a statistically significant decrease in PDK1 concentration (*p* < 0.05).

In the case of AKT1/2, immunofluorescence is reduced following the augmentation of NP-427 concentrations (Figure 8A–D). Similar to the anterior cases, the Western blot showed a decrease in phospho-Thr308-Akt, phospho-Ser473-Akt, and total AKT at 24 h in cells treated with NP-427 at 0.5 and 0.75 mg/mL (*p* < 0.05) (Figure 8E).

### 3.7. PHT427-Loaded NPs Increase Levels of Superoxide Anions

Fluorescence for the oxidized dihydroethidium (DHE) probe was used to detect superoxide anion (O^2−^) by visualizing the increased intensity of red fluorescence in the nuclei of cells. FaDu cell cultures in the NP-427 group (Figure 9A) show an increase in fluorescence in the cell nucleus regarding control and NP-Ø groups. Statistical significance was obtained by analyzing fluorescence using ImageJ software; thus, NP-427 showed an DHE intensity of 21.6 ± 0.6 (at 0.5 mg/mL) and 23.8 ± 0.5 (at 0.75 mg/mL) arbitrary units (AU) (*p* < 0.05) in comparison with the control group, which showed 16.7 ± 0.4, and with the NP-Ø group, which showed 17.1 ± 0.5 AU (Figure 9B).

### 3.8. EGFR Decreases in the Presence of NPs Loaded with PHT-427

The EGFR levels were measured by Western blot. The results show that EGFR levels in FaDu cells after the treatment at 24 h were lower in the case of NP-Ø and NP-427 when compared with the control (Figure 10). The EGFR levels were significantly (*p* < 0.05) decreased using NP-427 concerning the control and NP-Ø. Particularly, the EGFR expression decreased by 1981 ± 12 and 1211 ± 18 AU at concentrations of 0.5 mg/mL and 0.75 mg/mL of NP-427, respectively, concerning the control (3610 ± 11 AU).

## 4. Discussion

Targeted therapy using inhibitors has led to a new paradigm in the treatment of cancer. Overexpression or mutation of the PI3K signaling pathway has been identified as one of the key regulators in several malignancies, including HNSCC [5,6,7,8]. PHT-427 is a molecule inhibitor-targeting PI3K/AKT pathway with promising antitumor activity in various cancers but is not studied at HNSCC [11,12]. Therefore, this is the first study to investigate the effect of PHT-427 therapy on FaDu (HNSCC) cells.

In our study, PHT-427 was chosen as a competitive inhibitor derived from sulphonamides that acts specifically at two sites, AKT and PDK1, in the PI3K pathway at the same time [12,24]. These two sites are related to a PH domain involved in the PIP3-AKT joint and phosphorylation of Threonine 308 from AKT by PDK1. This inhibitor is used in research in different tumors such as in the pancreas [25], prostate [26], or skin [27], but as a hydrophobic drug and insoluble in an aqueous medium, its bioavailability is limited to oral formulation. Therefore, more efficient methods are required to improve the delivery of PHT-427 to cancer cells

Polymeric nanoparticles are promising vehicles to transport chemotherapeutic drugs due to their high versatility, easy surface functionalization, and modulation of physicochemical properties by changing the polymer composition and microstructure [14,15,16,17,18]. In the present preclinical study, we investigated the employment of the poly(VP-*co*-MTOS) nanoparticle drug delivery system, developed by our group, to improve the therapeutic effect of PHT-427 for the treatment of HNSCC. Previously, we successfully used this delivery system to transport α-TOS (an antitumor drug) in the treatment of HNSCC [20,21]. The present study shows the efficacy of PHT-427 loaded in poly(VP-*co*-MTOS) nanoparticles in HNSCC, demonstrating antitumoral activity.

Treatment with NP-427 was confirmed to significantly decrease the viability and to increase apoptosis of FaDu cells. NP-427 concentrations between 1.0 mg/mL and 0.5 mg/mL inhibited the FaDu viability below 70% during the first 24–48 h (Figure 2 and Figure 3). The data we obtained on cell viability are related to previous studies of Lucero-Acuña et al. [19]. They encapsulated the inhibitor PHT-427 in poly (lactic-co-glycolic acid) nanoparticles (PLGA) to treat BxPC3 and MiaPaCa-2 cells (pancreatic cancer cells). In cell viability tests that assumed the biphasic release behavior of PHT-427, the results showed that encapsulating in PLGA decreased the cell viability of MiaPaCa-2 and BxPC3 cells relative to treatment with PHT-427 alone. However, when they considered an alternative assumption that the PHT-427-PGLA rapidly delivered its entire contents to the cytosol soon after cell internalization (not biphasic release), the results showed that the encapsulation did not improve the therapeutic efficacy against the BxPC3 and MiaPaCa-2 cells. Moreover, these effects on cell viability were studied in periods of 1 to 7 days against our tests, where we saw effects from 24 h. Previous studies of Tian et al. with the inhibitor BEZ235 (dual inhibitor of PI3K-mTOR) loaded in delivery systems, such as liposomes, significantly reduced the cell viability of the HN5 head and neck squamous cell carcinoma cell line compared with the control and inhibitor alone at 72 h [28]. The increase in PI3K/Akt overexpression of up to 50% of all HNSCC cases suggests that inhibition of the PI3K/Akt/PDK1 signaling pathway is an effective HNSCC cancer therapy [29,30]. Changes in gene expression, protein levels, and immunofluorescence patterns illustrate the inhibition to the PI3K/AKT/PDK1 pathway implicated in cell growth and survival (Figure 5, Figure 6, Figure 7 and Figure 8). A gene expression study is crucial to understand how the inhibitor affects cell growth and promotes apoptosis. Our in vitro results demonstrated that NP-427, compared with the control and unloaded-nanoparticle groups, significantly suppress the gene expressions of PI3KCA, PDK1, and AKT (Figure 5). At the protein level, immunofluorescence patterns and Western blot data illustrate the decrease in markers such as PI3K, p-PDK1-Ser241, Total AKT, p-AKT-Ser473, and p-AKT-Thr308 (Figure 6, Figure 7 and Figure 8). This decrease in markers coincides with a decrease in the cell population, gene expression, and the induction of apoptosis in FaDu cells (Figure 2, Figure 3 and Figure 5).

The tensin homolog (PTEN), a tumor suppressor and phosphatase in normal physiology, regulates PI3K. However, the loss or inactivation of PTEN is a frequent alteration in cancer, leading to hyperactivity of the PI3K pathway [6]. Several inhibitors have been developed to inhibit this pathway for HNSCC, in combination or not with chemotherapy, immunotherapy, or radiation [10,31,32]. BYL719, an inhibitor of the PI3K pathway, concomitant tested with Dacomitinib [33], demonstrates synergic action against HNSCC cells with PI3CA mutation. In another study, adenosine induces intrinsic apoptosis of HNSCC cells via the PI3CA/AKT/mTOR signaling pathway, decreasing the phosphorylation levels of PI3K, AKT, and mTOR [34]. Herzog and colleagues [35] studied double inhibition on PI3K/AKT/mTOR by oral intake of PF-04691502. This tumor suppressor acts as a competitive inhibitor of ATP in PI3K and mTOR. To date, there are no other studies that analyze the PHT-427 inhibitor in HNSCC. Finally, PHT-427 has also been studied as a stabilizer of AKT dysregulation in skin cancer promoted for Rapamicine [27]. Our results are in agreement with the antitumor effect observed in the study by Kobes et al. [25]. This research sought to improve the treatment of pancreatic cancer via the drug delivery of inhibitor PHT-427 from PLGA nanoparticles. PLGA-PHT-427 showed the elimination of primary pancreatic tumor in 68% of mice and a six-fold to four-fold reduction in tumor volume relative to untreated tumors.

One extensively studied receptor tyrosine kinase upstream of PI3K/Akt signaling is EGFR. Overexpression of EGFR has been identified in many cancers from epithelial origins, including HNSCC, where overexpression of EGFR is found in over 95% of all tumors [25,26] and has been associated with a more aggressive malignant phenotype, including increased resistance to treatment and poorer clinical outcome. The findings of the current study show that PHT-427-loaded nanoparticle decrease EGFR levels in the FaDu cell line (Figure 10). This is comparable with a previous report of BYL719 (a specific inhibitor for PI3K p110α) in human head and neck cancer cell lines.

Oxidation in tumoral cells is important to maintain homeostasis. Normally, tumoral cells tend to endure high levels of oxidation to maintain a considerable division rate. For this reason, mitochondrial activity is increased and transmembrane proteins are capable of ejecting the overflow of superoxide anion out of the cell. In cases with an overload of superoxide anion in cells, homeostasis mechanisms may not be enough and apoptosis could be triggered, even in tumoral cells [36]. DHE, for superoxide anion detection, shows an increase in oxidation in cells treated with NPs (red color intensified) most clearly seen in cells with 0.75 mg/mL (Figure 9). The MTOS included in the NP structure promotes an elevation in cell oxidation and may help the apoptotic properties of PHT-427 itself, as our group demonstrated before [20,21].

## 5. Conclusions

The inhibitor PHT-427 is a promising targeted antitumor agent that has shown strong effects against several cancers but has not been tested in HNSCC. Our research in vitro shows the possibility to use this AKT/PDK1 pathway in new treatments of HNSCC, especially with the PHT-427 inhibitor. By utilizing NPs, we can resolve the problem of the administration and hydrophobic properties of PHT-427. However, an anticancer evaluation of the PHT-427-loaded MTOS-nanoparticles in animal models is still needed. Further preclinical and clinical studies of PHT-427-loaded nanoparticles alone or in combination with chemotherapeutic drugs could be beneficial to cancer patients.

## Figures and Tables

**Figure 1 pharmaceutics-13-01242-f001:**
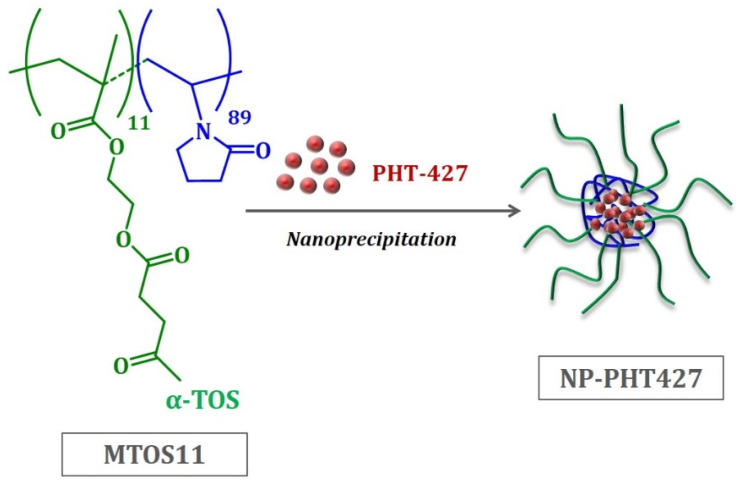
Representative scheme of the formation and the structure of NP-PHT427.

**Figure 2 pharmaceutics-13-01242-f002:**
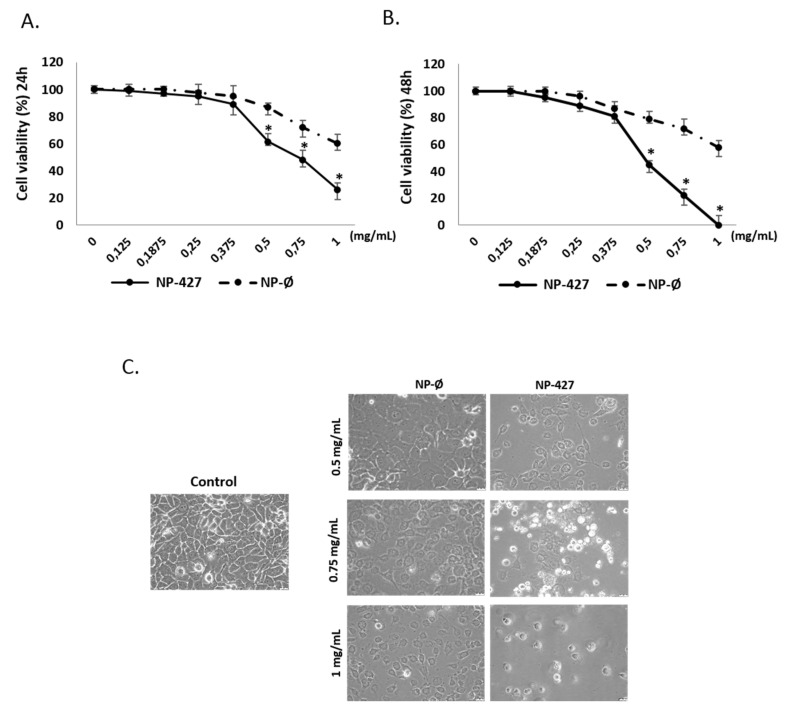
Inhibition of the proliferation of FaDu cells treated with NP-427. The FaDu viability of cultures measured by Alamar blue at 24 (**A**) and 48 (**B**) hours treated with NP-Ø (unloaded NPs) at 0, 0.125, 0.1875, 0.25, 0.375, 0.5, 0.75, and 1 mg/mL and NP-427 (PHT427-loaded NPs) at 0, 0.125, 0.1875, 0.25, 0.375, 0.5, 0.75, and 1 mg/mL. The curves include the mean, the standard deviation (*n* = 3), and the analysis of variance (ANOVA) results (* *p* < 0.05 statistically significant difference with vs. NP-Ø group). (**C**) Representative optical micrographs of FaDu cells at 24 h of treatment (40×). Scale bar 100 μm.

**Figure 3 pharmaceutics-13-01242-f003:**
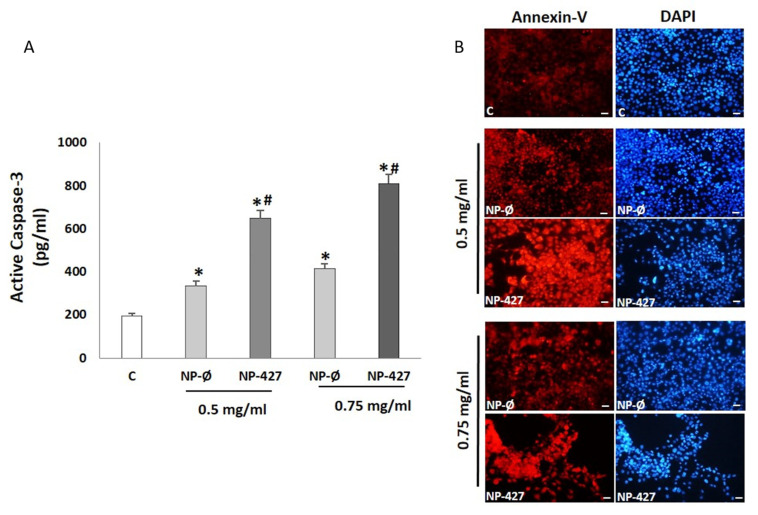
NP-427 increase the production of apoptosis in FaDu cells. (**A**) Caspase-3 in FaDu cultures treated with: C (control), NP-Ø (unloaded NPs, 0.5 and 0.75 mg/mL), and NP-427 (PHT427-loaded NPs, 0.5 and 0.75 mg/mL) for 24 h. The diagrams include the mean ± SD (pg/mL) (*n* = 4). * *p* < 0.05 versus control group and ^#^
*p* < 0.05 versus NP-Ø group. (**B**) Representative fluorescence micrographs of the immunostaining of Annexin-V (red fluorescence) in FaDu cultures treated with C (control), NP-Ø (unloaded NPs, 0.5 and 0.75 mg/mL), and NP-427 (PHT427-loaded NPs, 0.5 and 0.75 mg/mL) for 24 h. Nuclei (blue fluorescence) were identified by DAPI. Scale bar: 100 μm.

**Figure 4 pharmaceutics-13-01242-f004:**
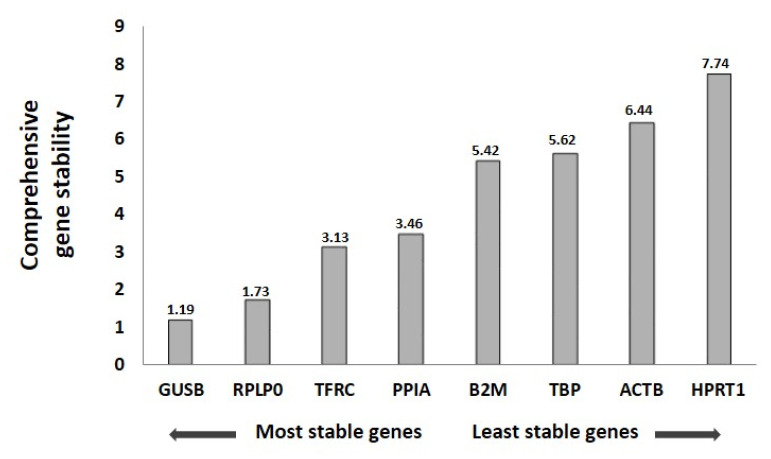
RefFinder ranking of the stability of eight reference genes tested for qRT-PCR analysis in FaDu. FaDu cultures treated with C (control), NP-Ø (unloaded NPs, 0.5 and 0.75 mg/mL), and NP-PHT427 (PHT427-loaded NPs, 0.5 and 0.75 mg/mL) for 24 h.

**Figure 5 pharmaceutics-13-01242-f005:**
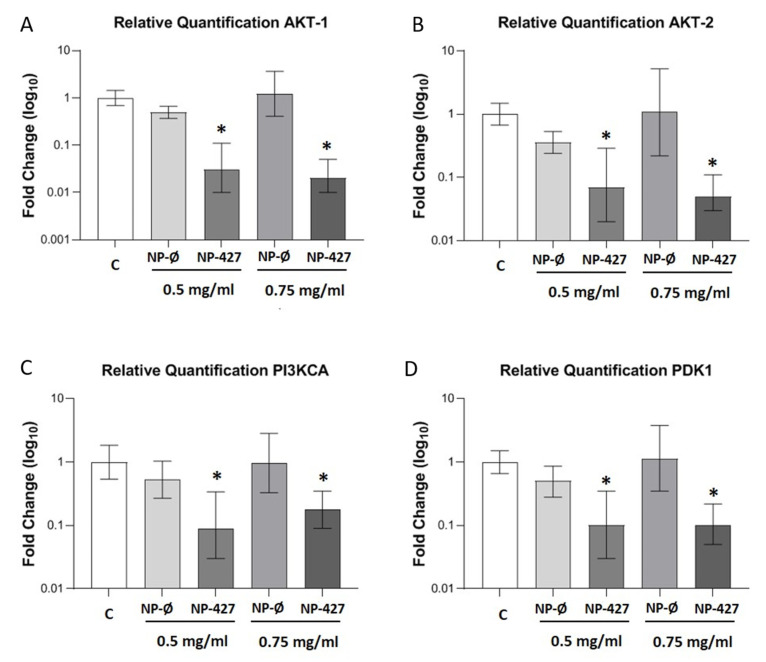
PHT-427-loaded NPs reduced the gene expression of the PI3K pathway in FaDu cells. Gene expression of (**A**) AKT1, (**B**) AKT2, (**C**) PI3KCA, and (**D**) PDK1 in FaDu cultures treated with C (control), NP-Ø (unloaded NPs, 0.5 and 0.75 mg/mL), and NP-PHT427 (PHT427-loaded NPs, 0.5 and 0.75 mg/mL) for 24 h. The diagrams include the mean ± SD Fold change (log_10_) (*n* = 4). * *p* < 0.05 versus control and NP-Ø groups. Reference genes: GUSB and RPLP0.

**Figure 6 pharmaceutics-13-01242-f006:**
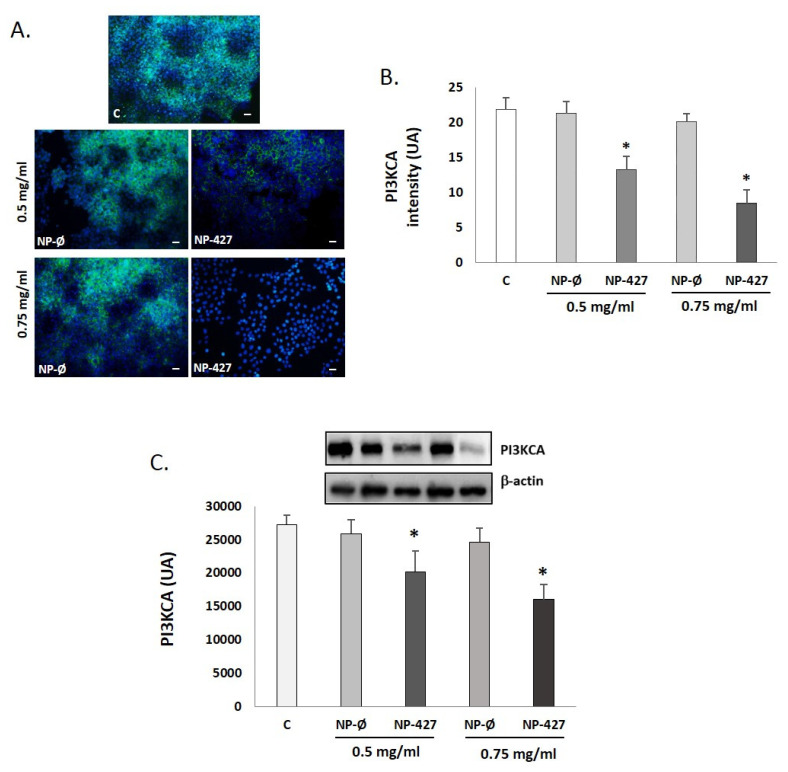
PI3KCA levels decrease with the treatment of nanoparticles loaded with PHT-427 on FaDu cells. (**A**) Representative Figure 3. KCA (green fluorescence) in FaDu cultures treated with C (control), NP-Ø (unloaded NPs, 0.5 and 0.75 mg/mL), and NP-427 (PHT427-loaded NPs, 0.5 and 0.75 mg/mL) for 24 h. DAPI labels nuclei with blue fluorescence. Scale bar: 100 μm. The images were taken at random in a blinded fashion. (**B**) The bar graphs show the immunofluorescence intensity changes for PI3KCA by Image J. The values represent the mean ± SD arbitrary units (AU) in triplicate. * *p* < 0.05 versus the control and NP-Ø groups (*n* = 4). (**C**) A representative Western blot and densitometry analysis for PI3KCA protein for each experimental group are shown. Loading control: β-actin. The bar diagrams represent the mean ± SD of arbitrary units (AU). * *p* < 0.05 vs. C and NP-Ø groups (*n* = 4).

**Figure 7 pharmaceutics-13-01242-f007:**
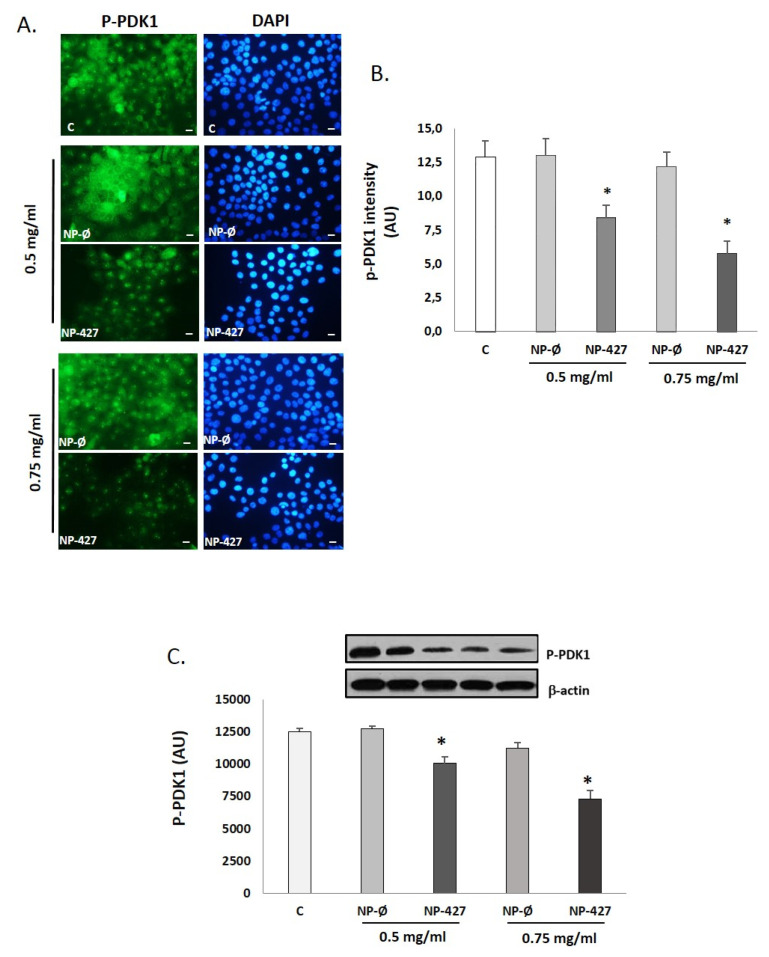
p-PDK1 levels decrease with the treatment of nanoparticles loaded with PHT-427 on FaDu cells. (**A**) Representative fluorescence images of the immunostaining of p-PDK1-Ser241 (green fluorescence) in FaDu cultures treated with C (control), NP-Ø (unloaded NPs, 0.5 and 0.75 mg/mL), and NP-427 (PHT427-loaded NPs, 0.5 mg/mL and 0.75 mg/mL) for 24 h. DAPI labels nuclei with blue fluorescence. Scale bar 100 μm. The images were taken at random in a blinded fashion. (**B**) The bar graphs show the immunofluorescence intensity changes for PI3KCA by Image J. (**C**) The p-PDK1-Ser241 protein levels were analyzed by Western blotting in FaDu cells. Densitometry analyses for each experimental group are shown. Loading control: β-actin. The diagrams represent the mean ± SD of arbitrary units (AU). * *p* < 0.05 vs. C and NP-Ø groups (*n* = 4).

**Figure 8 pharmaceutics-13-01242-f008:**
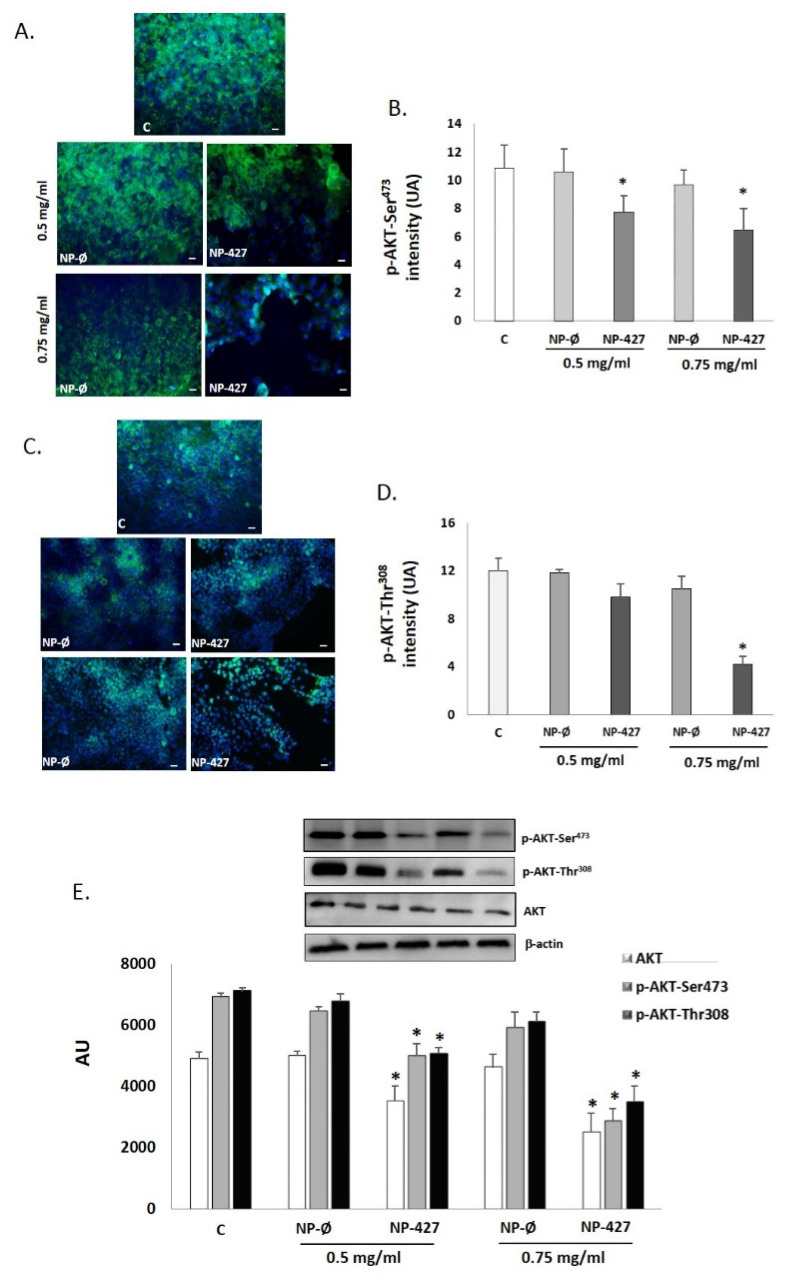
AKT levels decrease with the treatment of nanoparticles loaded with PHT-427 on FaDu cells. Representative fluorescence micrographs of the immunostaining of p-AKT-Ser^473^ (green fluorescence) (**A**) and p-AKT-Thr^308^ (green fluorescence) (**C**) in FaDu cultures treated with C (control), NP-Ø (unloaded NPs, 0.5 and 0.75 mg/mL), and NP-427 (PHT427-loaded NPs, 0.5 and 0.75 mg/mL) for 24 h. The nuclei were identified in blue fluorescence by DAPI. Scale bar: 100 μm. The images were taken at random in a blinded fashion. (**B**,**D**) ImageJ quantifies fluorescence intensity for p-AKT-Ser^473^ (**B**) and p-AKT-Thr^308^ (**D**). The values represent the mean ± SD arbitrary units (AU) in triplicate. * *p* < 0.05 versus control and NP-Ø groups (*n* = 4). (**E**) p-AKT-Thr^308^, p-AKT-Ser473, and total AKT protein levels were analyzed by Western blotting in FaDu cells protein extracts from the different treatment group. A representative Western blot and densitometry analysis for each protein are shown. β-actin was used as a loading control. The diagrams represent the mean ± SD of arbitrary units (AU). * *p* < 0.05 vs. control and NP-Ø groups (*n* = 4).

**Figure 9 pharmaceutics-13-01242-f009:**
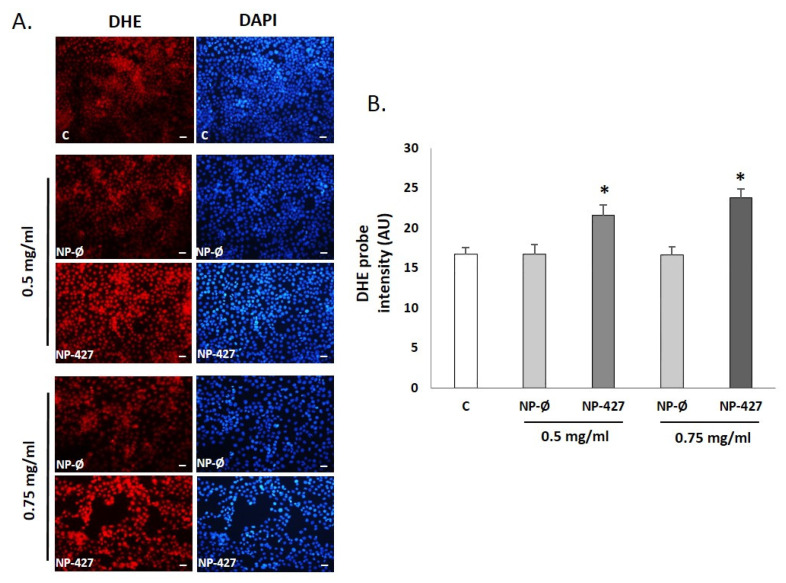
NP-427 increases the production of superoxide anion (O^2−^) in FaDu cells. (**A**) Representative images of fluorescence micrographs of the FaDu cells (×40). Scale bar 100 μm. (*n* = 4). Superoxide anion detection with the fluorescent probe dihydroethidium, DHE (red fluorescent), and DAPI (a marker of the cell nuclei with blue fluorescent) dye in FaDu cultures. Experimental groups: C (control), NP-Ø (unloaded NPs, 0.5 and 0.75 mg/mL), and NP-427 (PHT427-loaded NPs, 0.5 and 0.75 mg/mL) for 24 h. (**B**) Quantification of fluorescence intensity for the DHE probe by ImageJ. The diagrams represent the mean ± SD of arbitrary units (AU). * *p* < 0.05 vs. their respective NP-Ø and control group (*n* = 4).

**Figure 10 pharmaceutics-13-01242-f010:**
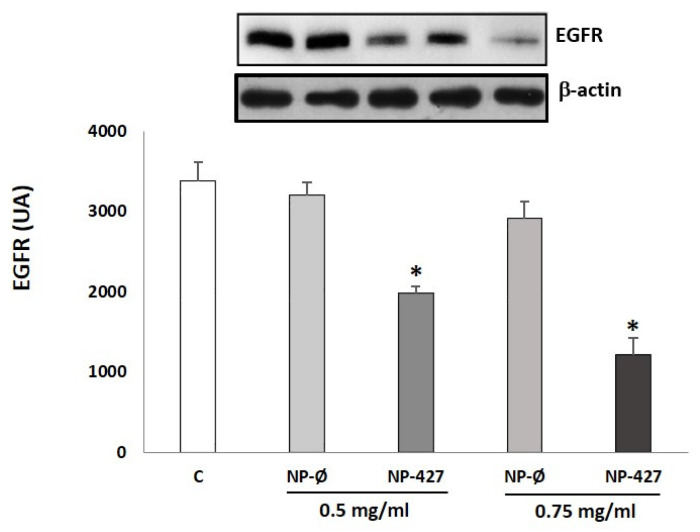
NP-427 reduces EGFR levels in the FaDu cell culture. FaDu cells treated with C (control), NP-Ø (unloaded NPs, 0.5 and 0.75 mg/mL), and NP-427 (PHT427-loaded NPs, 0.5 and 0.75 mg/mL) for 24 h. The densitometric data of Western blot for EGFR represent the mean ± SEM of arbitrary units (AU). Loading control: β-actin. * *p* < 0.05 vs. C and their respective NP-Ø group (*n* = 4).

**Table 1 pharmaceutics-13-01242-t001:** Most relevant characteristics of NP-Ø and NP-427: hydrodynamic diameter (Dh), polydispersity index (PDI), zeta potential (ξ), encapsulation efficiency (EE), loading capacity (LC), and encapsulated drug concentration. Values represent mean ± SD.

NP Sample	PHT-427 (%*w*/*w*)	PHT-427 (mg/mL)	*D*_h_ (nm)	PDI	ζ (mV)	EE (%)	LC (%)	Encapsulated PHT427 mg/mL
NP-Ø	0	---	132.1 ± 3.3	0.098 ± 0.010	−2.62 ± 0.2	---	---	---
NP-427	10	0.2	155.2 ± 9.7	0.124 ± 0.015	−3.45 ± 0.1	75.4	9.55	0.151

**Table 2 pharmaceutics-13-01242-t002:** Ranking order of eight reference genes tested for qRT-PCR studies in FaDu cells. FaDu cultures treated with C (control), NP-Ø (unloaded NPs, 0.5 and 0.75 mg/mL), and NP-PHT427 (PHT427-loaded NPs, 0.5 and 0.75 mg/mL) for 24 h.

Method	1	2	3	4	5	6	7	8
Delta CT	GUSB	TFRC	RPLP0	PPIA	TBP	B2M	ACTB	HPRT1
BestKeeper	RPLP0	GUSB	PPIA	B2M	ACTB	TFRC	HPRT1	TBP
Normfinder	GUSB	TFRC	RPLP0	PPIA	TBP	B2M	ACTB	HPRT1
Genorm	GUSB/RPLP0		PPIA	TFRC	TBP	B2M	ACTB	HPRT1
**Recommended comprehensive ranking**	**GUSB**	**RPLP0**	**TFRC**	**PPIA**	**B2M**	**TBP**	**ACTB**	**HPRT1**

## Data Availability

The data are available from the authors upon request.

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
