# Peer review of "Antitumor Activity of Nanoparticles Loaded with PHT-427, a Novel AKT/PDK1 Inhibitor, for the Treatment of Head and Neck Squamous Cell Carcinoma"

_pharmaceutics, 2021, doi:10.3390/pharmaceutics13081242_

Round 1

Reviewer 1 Report

Before be accepted, the following issues should be addressed:

  1. To further improve the quality of the manuscript and help readers to know nanomedicine for cancer therapy, some references on polymeric nanomedicine should be cited in section of introduction, such as Molecules, 23:826; Chinese Chemical Letters, 2019, 30:1083-1088; Chinese Chemical Letters, 2020, 31:3178-3182; Bioactive Materials, 2021, 6: 1513-1527.
  2. Although the polymer used in this study had been reported, it is better to show the structure and introduce the basic information of the poly(VP-co-MTOS), which will help readers to understand.
  3. The release profiles of PHT-427-loaded poly(VP-co-MTOS) NPs should be tested.
  4. Stability of nanomedicine is important for nanomedicine application in vivo. Thus, how is about the nanomedicine in manuscript?

Reviewer 2 Report

This paper introduces about the cancer therapy effect of PHT-427 loaded nanoparticles by observing the biomarker and cell viability.

  1. Other papers and researchers have demonstrated the effect of PHT-427 on several types of cancer cells and tissues. It is noted that this paper demonstrated the effectiveness of PHT-427 in HNSCC for the first time. However, the author does not explain reason that why the PHT-427 effect is difficult to prove in the HNSCC.
  2. PHT-427 is loaded into MtosNP and used, but there is not enough explanation for why PHT-427 should be loaded into MtosNP.
  3. The cell viability results of PHT-loaded MtosNP and bare MtosNP was introduced in the manuscript. To demonstrate a synergy effect of PHT-4276 loaded MtosNP, cytotoxicity results with the same amount of PHT-427 should be attached.
  4. The loading efficiency of PHT-427 is high at 75.4%. What interaction between np and ph-427 drives this phenomenon?
  5. In addition to encapsulation efficiency, loading capacity must also be calculated.

Reviewer 3 Report

In their manuscript, Yanes-Diaz et al. presented results obtained on the evaluation of the anti-tumor activity of nanoparticles loaded with PHT-427 for the treatment of head and neck squamous cell carcinoma. Globally, the presented results are interesting but the manuscript needs to be carefully read because several sentences need to be reworded. Moreover, some experimental data are missing and have to be added (see comments below).

In view of these general remarks and the following ones, I do recommend the publication of the manuscript in Pharmaceutics after major revision.

Please find below specific comments and questions that need to be addressed before any publication.

  1. Page 1, Abstract line 33: The abbreviation “NPs” has to be defined, even if its meaning is obvious.
  2. Page 2, lines 53-56: I don’t understand the sentence “Most abnormalities … suppressor PTEN [10]”. I guess some words are missing.
  3. Page 2, line 58: “Therefore, has …” A subject is missing in this sentence.
  4. Page 2, lines 60-62: What is/are the advantage(s) of the approach selected by the authors compared to the one(s) described in the literature or already used?
  5. Page 2, lines 70-72 “Polymeric nanoparticles … drug toxic effects.”: The authors have to give; at least, one reference on the use of nanoparticles as drug delivery systems: there are several recent reviews on the subject.
  6. Page 2, lines 75-77 “In this line, … nanoparticles (NPs).”: this sentence has to be reworded to be understandable.
  7. Page 2, line 79: What does “poly(VP-co-MTOS) (89:11) mean? If it is a block copolymer, the authors have to change to poly(VP)-block-poly(MTOS). What does (89:11) mean? Is it a weight percent? Is it the number of repeating units of each block?
  8. Page 2, lines 88-91: What does the number 11 after MTOS mean? Is 89:11 mean 89% and 11%? What are the molar mass and distribution of the synthesized copolymer?
  9. Page 2, line95: “72 hours”, It is a very long time for dialysis: encapsulated PHT-427 can be released from the nanoparticles. Why don’t the authors used faster methods to eliminate non-loaded PHT-727 such as ultra-centrifugation/filtration method?
  10. Page 4, line 157: I suggest to change “Then” to “After”.
  11. Page 4, lines 166-168: The sentence “Based on each program … overall classification [17].” Has to be reworded.
  12. Page 5, Table 1: The authors have to add standard deviation for the zeta potential values.
  13. Page 5, lines 215-218: The sentence “Particularly, … to ISO 10993-5: 2009).” needs to be reworded.
  14. Page 5, lines 219-224: The authors have to calculate the IC50 of their PHT-427 loaded-NPs from the graphs given in Figure 1A and Figure 1B. Moreover, how can they conclude that the higher toxicity observed for NP-427 after 48 hours of incubation is the result of PHT-427 release from NPs without studying PHT-427 release with time in various media? The PHT-427 release profile has to be added in their study.
  15. Page 5, line 231: I guess that the verb “obtained” is missing between “images” and “by optical”.
  16. Page 6, Figure 1: The authors have to add the units of each axes, as well as axis title. I guess that the y-axis corresponds to cell viability in percent. But what does the x-axis represent?

In the legend of this figure, the authors said that they treated FaDu cells with “C (control”. Where are the curves corresponding to those conditions on Figure 1A and Figure 1B?

Finally, the quality of images given in Figure 1C needs to be improved.

  1. Page 7, paragraph 3.4.: I don’t understand how the authors determined the given genes ranking. It might be interesting to explain more in details how they obtained data gathered in Table 2 and Figure 3.
  2. Page 13, lines 348-350: The sentence “Particularly … NP-427, respectively” needs to be reworded.
  3. Page 14, line 362: “a novel AKT/PDK1 specific inhibitor”: It is already mentioned just above on lines 360-361.
  4. Page 14, line369: I guess that “by” has to be changed to “to”.
  5. Page 14, lines 372-374: As mentioned in my comments on the Introduction part, the authors have to give at least one reference on NPs used as drug delivery systems.
  6. Page 14, lines 383: I don’t understand why the authors started the sentence with “About our study”. It has to be modified.
  7. Page 14, line 388: I guess that “PH-427” has to be changed to “PHT-427”. Same remark for “PH-427” mention on page 15 line 389.
  8. Pages 14-15, lines 382-396: All this paragraph needs to be reworded and the links with the authors study have to be more clearly explain.
  9. Page 16, page 447: I think that the comma between “Our research” and “in vitro” is not necessary.
  10. Page 16, lines 452-455: The sentence “Because limited in vivo … of cancer treatment.” Has to be reworded.

Reviewer 4 Report

The paper entitled “Antitumor activity of nanoparticles loaded with PHT-427, a novel AKT/PDK1 inhibitor, for the treatment of head and neck squamous cell carcinoma” by oaquín Yanes-Díaz , Raquel Palao-Suay , Maria Rosa Aguilar , Juan Ignacio Riestra-Ayora , Antonio Ferruelo-Alonso , Luis Rojo del Olmo , Blanca Vázquez-Lasa , Ricardo Sanz-Fernandez , Carolina Sánchez-Rodríguez presents a study concerning the synthesis, physical-chemical characterization, and evaluation the antitumoral effect of PHT-427-loaded polymeric nanoparticles based on block copolymers of N-vinyl pyrrolidone and a methacrylic derivative of α-tocopheryl succinate. The topic of the work is quite interesting and well written, there is one point to address before publication:

-    Please add SEM or TEM pictures of your NPs.

Round 2

Reviewer 2 Report

The revised manuscript is good for publication in the pharmaceutics.